# Physiologic, Genetic and Epigenetic Determinants of Water Deficit Tolerance in Fruit Trees

**DOI:** 10.3390/plants14121769

**Published:** 2025-06-10

**Authors:** Marie Bonnin, Khadidiatou Diop, Gabriel Cavelier, Mathieu Crastes, Renel Groenewald, Hong Thu Nguyen, Raphaël Morillon, Frédéric Pontvianne

**Affiliations:** 1Plant Genome and Development (LGDP), The French National Centre for Scientific Research (CNRS), 66860 Perpignan, France; khadidiatou.diop@univ-perp.fr; 2UMR AGAP, 34398 Montpellier, France; raphael.morillon@cirad.fr; 3Functional Biology and Ecology, TULIP Graduate School, 31062 Toulouse, Francegroenewald08@gmail.com (R.G.);; 4University of Toulouse III—Paul Sabatier, 31062 Toulouse, France; 5University of Perpignan Via Domitia (UPVD), 66860 Perpignan, France

**Keywords:** fruit crop, genome-wide association studies, drought stress, gene editing, polyploidy, gene regulation, epigenetic modification

## Abstract

Fruits are increasingly recognized as an important part of a healthy diet. Fruit crops represent a wide range of woody perennial species grown in orchards. Water availability is a primary environmental factor limiting fruit crop growth and productivity. Erratic rainfall patterns and increased temperatures due to climate change are likely to increase the duration of droughts. This review aims to highlight the different mechanisms by which fruit crops respond to water stress deficits. Emphasis is placed on physiological, genetic and epigenetic determinants of stress response in fruit crops. These findings can contribute to a deeper understanding of the underlying effects of drought. We also describe new research opportunities made possible by the increasing availability of population-level genomic data from the field, including genome-wide association studies (GWAS) and high-throughput phenotyping.

## 1. Introduction

Drought is one of the most severe environmental constraints worldwide, and thus it has become a major concern for agricultural stakeholders. Water shortage impact on crops impedes plant growth and can cause a decrease crop productivity. Fruit tree growth stages are highly sensitive to water availability and temperature. With climate change and global warming, fruit crops are exposed to increased temperature and drought [1]. Breeding programs are increasingly designed to improve fruit crop tolerance to drought stress. However, breeding new genotypes of perennial fruit crops is slow due to extended breeding cycles, juvenility leading to prolonged waits for flowering, and genetic heterogeneity [2]. Understanding how fruit crops in orchards respond to drought but also identifying genotypes that are more adapted to drought appear to be more and more necessary.

Plants have evolved displaying a variety of physiological and biochemical responses at cellular and whole-organism level to tolerate drought stress, thus making it a complex phenomenon. This review gives an overview of recent research into the physiological, genetic, and epigenetic responses to drought stress of major fruit crops. Here we consider that a perennial fruit crop is “major” when it is of economic interest and largely cultivated (apple, citrus, apricot, peach, pear and sherry trees for example) worldwide. We aim to answer the following question: how can we move towards a more integrative definition (physiological, biochemical, genomical and epigenomical) of fruits crops drought tolerance?

## 2. Fruit Crops Combating Water Scarcity: Physiological and Biochemical Responses and Regulatory Pathways

### 2.1. The Main Effects of Water Stress on Crops

Transpiration is caused by the evaporation of water at the leaf–atmosphere interface; it creates negative pressure (tension) at the leaf surface and creates in the leaf a water potential gradient between these cells and their neighbors, which is transmitted from close cells, due to the cohesion of water molecules [3]. Water uptake in plants is possible if the cellular water potential remains lower than that of the extracellular medium and the soil. In fact, the transpiration-induced decrease in water potential in the stomatic chamber creates a water potential gradient between these cells and their neighbors (Figure 1).

This gradient provides the force needed to move water around the plant. In this way, water passes from the soil to the roots. Water passes then from the root tissues through symplastic and apoplastic routes to the conductive xylem tissues, crossing the Casparian strips associated with suberin lamellae [4]. These different barriers, in association with transmembrane water channels (aquaporins) regulate water flow. When soil osmotic pressure rises, for example under water shortage conditions, roots react by increasing their production of abscisic acid. When abscisic acid is perceived by the leaves, stomata close to limit water loss through transpiration, which further contributes to limit water uptake by roots (Figure 1). Plants can also produce compatible solutes, or osmoprotectants, to restore the higher water potential of root cells and thus encourage the passage of water. In apple (Malus domestica), root xylem vessel formation changes to have a greater hydraulic conductivity, increasing drought tolerance [5].

### 2.2. Molecular and Metabolic Adjustments to Water Scarcity

Transcriptomics makes it possible to analyze the evolution of gene expression in response to environmental changes [6,7]. Thus, it offers a global view of plant genomes in each situation [8]. Gene expression analysis is crucial in research on plants of agronomic interest [9]. This method plays a key role in identifying genes useful for improving traits related to production, reproduction, fruit quality or stress tolerance, particularly in citrus [10,11,12]. Drought stress triggers significant alterations of the citrus transcriptome, leading to the up-regulation or down-regulation of numerous genes. Deregulated genes are mostly associated with plant defense responses, photosynthesis, cell wall modification, and metabolism [13]. Up-regulation of antioxidant response related genes suggests a physiological response to drought stress, while the downregulation of photosynthetic genes highlights the detrimental effects of this stress on the primary metabolic processes. Exposure to drought, leads to progressive removal of water from the cytoplasm. Decrease of cytosolic and vacuolar volumes triggers production of specific sets of primary and secondary metabolites that act as osmoprotectants, antioxidants, and stress signals.

**Figure 1 plants-14-01769-f001:**
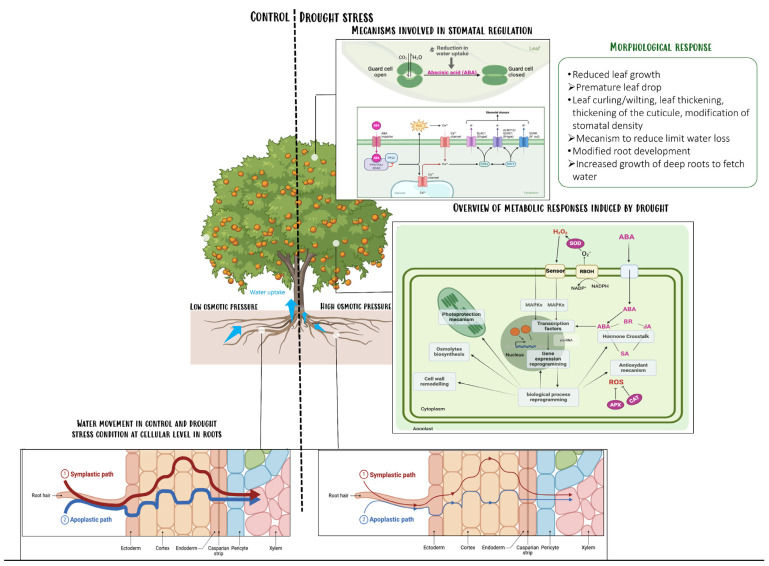
Schematic representation of drought main effects in trees. Transpiration stream from the leaves provides the force needed to move water around the plant [14]. At cellular level, in symplastic pathway, aquaporines may regulate the water flux (green box). In this way, water is transferred from the soil to the roots. In stress condition (water shortage), roots react by increasing their production of abscisic acid (ABA). When abscisic acid is perceived by the leaves, the stomata close to limit water loss through transpiration, which further contributes to limit water [15,16]. Detection of stress by wall sensors leads to reprogramming of gene expression. (part of the design was made with Biorender software, Created with BioRender.com).

#### Photosynthesis

In leaves, drought causes the closure of stomata, which are formed by paired guard cells that exchange gas in leaves, resulting in a significant decrease in net photosynthetic rates. A study on citrus showed that, under drought conditions, plants reduce stomatal conductance, leading to lower photosynthetic and transpiration rates [17]. Additionally, early stomatal closure helps reduce cavitation events and prevent xylem embolism, while more severe drought conditions induce cavitation events, maintaining embolism levels below 50% [18]. Genes that regulate water loss and CO₂ uptake can contribute to this phenomenon. For instance, ABA-responsive genes (e.g., NCED3, AREB1, PP2C, SnRK2): Involved in abscisic acid (ABA) signaling, which controls stomatal closure [18,19]. Aquaporins (PIP1, PIP2, TIPs) can also play an important role, by regulating water movement across membranes and influence mesophyll conductance [20]. The decrease in photosynthesis induced by water deficit could be linked to stomatal and/or non-stomatal factors [21]. Water shortage stress has direct and indirect effects on the chlorophyll content and photosynthetic efficiency of plants. Direct effects involve the regulation of the activity and expression levels of enzymes involved in chlorophyll biosynthesis and photosynthesis. Indirect effects concern specific regulatory pathways such as antioxidant enzyme systems [22]. Prolonged drought stress could lead to chlorophyll degradation and leaf senescence. Rapid chlorophyll breakdown mitigates the photo-oxidative toxicity of free chlorophyll and its colored catabolites [23]. This process is thought to protect plant cells from recycle nitrogen from Chl-binding proteins during leaf senescence. Several enzymes are involved. The early stage of degradation can be induced by stress conditions to initiate leaf senescence. Here, chlorophyllase (CLH) and Pheophytin Pheophorbide Hydrolase (Pheophytinase or PPH) play a role in breaking down Chl molecules [23]. Highly conserved among plant rice STAY GREEN (SGR), delays senescence and maintains photosynthetic activity. Several studies also pointed out that tomato (GREEN-FRESH), pepper (CHLOROPHYLL RETAINER) and pea (Mendel’s green cotyledon gene) cause retention of chlorophyll during leaf senescence despite a decrease in leaf functionality [23]. Chl breakdown initiation require SGR Protein interacting with both Chl catabolic enzymes and LHCII [24]. In plants, NYC1 (Non-Yellow Coloring 1) and NOL (NYC1-like) chlorophyll b reductase are also in charge of catalyzing the degradation of chlorophyll and maintaining the stability of the photosystem [25,26]. Stomatal closure in response to osmotic stress occurs more rapidly than the decrease in photosynthesis. By reducing water availability in the root zone, water deficit induces stomatal closure (hydrostatic closure), leading to a reduction in CO_2_ uptake and inhibition of carbon fixation [27]. Reduced CO_2_ uptake by Rubisco (EC 4.1) slows down the Calvin cycle [28,29,30]. Some genes have been identified to ensure efficient RUBISCO activity under drought conditions [31]. Despite the inhibition of the Calvin cycle, photochemical reactions continue to take place. Reduced NADP^+^ utilization (oxidized form) leads to its accumulation, and the electron transport chain (ETC) is saturated [32]. Excessive reduction of ETC leads to the production of ROS [33]. Photons captured by photosystems I and II (LHC I and II) generate electrons (e^-^) and a proton (H^+^) gradient, triggering the electron transport chain at photosystems (PS) I and II (via photolysis of H_2_O) and the production of NADPH and ATP by NADPH reductase and ATP synthase, respectively. In citrus it has been demonstrated that molecular response to drought could involve regulation of D1 protein (encoded by the psbA transcripts), essential for repairing PSII damage [34]. By regulating antenna complex stability under stress, LHCB (Light-Harvesting Complex B) genes are also thought to be involved in drought stress mitigation in in Peach (*Prunus persica L*.) [35]. However, excessive photon energy at LHC II converts the chlorophyll molecule (Chl) into an excited triplet form (^3^Chl*) that reduces O_2_ to ^1^O_2_. The reduced activity of the Calvin cycle, due to low CO_2_ content, leads to over-reduction of ETC, resulting in electron leakage. The over-reduction of QB and QA also directly reduces O_2_ to O_2_^--^. At the PSI, over-reduction of ETC causes electron leakage from ferredoxin (Fd) to O_2_, forming O_2_^--^ via the Mehler reaction. The O2^-^ generated is eliminated either spontaneously, or by the action of superoxide dismutase (SOD) to form H_2_O_2_ which, in the presence of reduced redox metals (Fe^2+^, Cu^+^), is transformed into a highly toxic OH^−^. The damage caused results in photoinhibition. Increased intracellular ROS concentration damages photosynthetic machinery proteins [36], inhibits the PSII repair machinery [37] and initiates programmed cell death in stressed cells [38].

Cellular water regulation

Plant cell membranes are packed with aquaporins that regulate the hydraulic conductivity of the cell membranes [39,40]. Aquaporins fall into five subfamilies categorized as Plasma membrane Intrinsic Proteins (PIPs), Tonoplast Intrinsic Proteins (TIPs), NOD26-like intrinsic proteins (NIPs), small basic intrinsic proteins (SIPs), and uncharacterized intrinsic proteins (XIPs) [41]. Diverse functions such as transport of water, gases like carbon dioxide and oxygen, uncharged molecules such as urea and glycerol, nutrient/metal/mineral ions, organic molecules and signaling molecules (hydrogen peroxide, H2O2) have been attributed to these aquaporin isoforms. Extensive work has been performed on model plants / annual plants to decipher the role of aquaporins to regulate the cellular membranes water permeability in response to cellular signals/environmental stresses [42]. In perennial fruit species, investigations regarding the function of aquaporins in water deficit conditions are still limited. Different works showed that the heterologous expression of apple PIP aquaporin (MdPIP1;3, MpPIP2;1) in model plants enhanced the drought tolerance [43,44]. In citrus roots, water-deficit tolerance is mediated by the down regulation of PIP gene expression accompanied by decreased plant vigour, thereby facilitating water retention in the cells under water stress conditions [45]. Modulation of the expression MIPs and TIPs were also observed in water deficit condition depending on the sensitivity of the studied genotypes [46,47]. In pear tree subjected to drought PIP1:4 and 2:6/2:7 genes of wild vs. cultivated pear tree species showed also modulation [48]. In Prunus, depending on the genotype, combined foliar sprays of Ca+B and Ca+Si, modulated the water and turgor potential in fruits [49]. Physical changes were associated with increases in some aquaporins in sherry and nectarine but not in apricot and doughnut peach [49]. In sweet sherry, PaPIP1;4 was shown to be upregulated by pre-harvest calcium treatments that is preventing cracking. In grapevine, a cysteine-rich peptide that induces stomata formation (i.e Epidermal Patterning Factor Like 9 (EPFL9), also known as STOMAGEN) was knockout through the CRISPR-Cas9 system. This resulted in the reduction of stomatal density in edited plants. Lower stomatal density allows decreasing the use of irrigation water and increased crop water-use efficiency [50].

A study on grapevine showed the influence of rootstock genotypes in the adaptive response of scions to water limiting conditions [51,52]. These authors also showed that scions could influence stomatal control of water transpiration. Cell-to-cell component of plant water transport in both rootstock and scion contributed to limit embolisms formation in roots and on hydraulics of leaves. Aquaporins plays a crucial role in rootstock to scion water flux. They also play an important role in regulating the flow of water within organs and between tissues. Especially in drought conditions. Aquaporins are involved in root hydraulic conductivity and water uptake, as well as in cellular osmoregulation. A study involving the more vigorous and drought-tolerant rootstock grapevine under water stress conditions was recently published by Labarga et al., 2023 [53]. They showed that, in the absence of stress, vines grafted onto 1103P and R110 rootstocks (the more vigorous and drought-tolerant) had higher photosynthesis, stomatal conductance (*g_s_*), and hydraulic conductance compared with the less vigorous and drought-sensitive rootstock (161-49C). In water stress condition, however, there were almost no differences between vines depending on the rootstock grafted. Interestingly, VvPIP and VvTIP aquaporins were up-regulated in the vines grafted onto 1103P and down-regulated in the ones grafted onto 161-49C. The authors concluded that better 1103P capability to tolerate drought was the result of up-regulation of all VvPIP and VvTIP aquaporins, lower ABA synthesis, and higher ACC/ABA ratios in leaves under stress conditions.

Osmoprotectants

Accumulation of osmoprotectants contribute to slower the cellular osmotic potential and draws water into the cell to maintain turgor pressure (Figure 1). Osmoprotectants protect the cellular apparatus from dehydration damages, without interfering with the normal metabolic processes at the cellular level. Such molecules include for instance amines (polyamines and glycine betaine), amino acids (proline), soluble sugars (glucose, sucrose, trehalose), and polyols (mannitol, sorbitol and inositol). Many fruit trees belong to the Rosaceae family. This family include for instance, trees of the genus Prunus, like sherry (*Prunus avium*) apricot (*Prunus armeniaca*), almond (*Prunus dulcis*), peach (*Prunus persica*), plum (*Prunus domestica*), Cydonia trees (*Cydonia oblonga*), shrubs of the Rubus genus like raspberry (*Rubus idaeus*) and blackberry (*Rubus fruticosus agg.*), trees of the genus Pyrus like pear (*Pyrus communis*) and trees of the Malus genus like apple (*Malus domestica*). Trees belonging to this family share a common mechanism in response to drought stress. They are known to produce Sorbitol in leaves as osmotic adjustment. Synthesis of sorbitol involves catalysis of glucose via aldose-6-phosphate reductase (A6PR), also called sorbitol-6-phosphate dehydrogenase (A6PR) (also called NADP-dependent sorbitol-6-phosphate dehydrogenase (S6PDH)). In apple and peach, it has been showed that water stress could induce sorbitol accumulation in drought-sensitive genotypes [54]. Apples genotypes with higher sorbitol levels were less susceptible to drought stress symptoms [55]. Studies on water stress have also demonstrated the involvement of LEA proteins in osmoprotection processes and in the adaptive response to stress [56,57]. They are hydrophilic proteins that play a role in maintaining membrane and protein structures [58]. In citrus leaves, expression of the cLEA V gene encoding the cLEA V protein is induced by oxidative, salt and water stress and high temperatures [59].

Antioxidants

Reactive oxygen species (ROS) are highly oxidizing compounds that damage cell integrity [60]. Environmental stress-induced stomatal closure can lead to a decrease in intracellular CO_2_ concentration (Ci) and the formation of singlet oxygen (^1^O_2_) [61]. A study on apple reported that MsDREB6.2 gene in the AP2/ERF family could play a role in stomatal closure and ROS scavenging, increasing tolerance under drought stress [62]. To counter the oxidative damage induced by salt stress, enzymes and non-enzymatic antioxidant compounds are involved in ROS detoxification. Oxidative stress in plants leads to the induction of several enzymes, including superoxide dismutase (SOD) (EC 1.15.1.1), catalase (CAT) (EC 1.11.1.6), ascorbate peroxidase (APX) (EC 1.11.1.11) and glutathione reductase (GR) (EC 1.6.4.2) (Figure 1). The first three are responsible for the conversion of superoxide ion to hydrogen peroxide (H_2_O_2_) and its subsequent reduction to H_2_O. The fourth is involved in ascorbate recycling. Removal of the O_2_^--^ radical occurs in two stages. Firstly, SOD converts the O_2_^-^ radical into H_2_O_2_ [63] (Mittler, 2002). Numerous publications demonstrate the induction of SOD under stress conditions, in particular water and salt stress. SOD, which catalyzes the elimination of superoxide radicals, can be considered the first line of defense against ROS. The H_2_O_2_ formed by SOD can then be converted to oxygen and H_2_O by CAT. Catalase is a tetrametric heme protein. It plays an important role in the antioxidant defense of plant cells under stressful conditions. Alternatively, H_2_O_2_ can also diffuse rapidly across membranes via aquaporins to reach other compartments and join the ascorbate-glutathione cycle. The oxidation of glutathione and ascorbate leads to the elimination of H_2_O_2_. In the first reaction, catalyzed by APX, ascorbate acts as a reducing agent and oxidizes to monodehydroascorbate (MDHA). Ascorbate peroxidase is a metallo-enzyme. Due to its higher affinity for H_2_O_2_ and its cellular ubiquity, it is considered one of the most important enzymes for the ROS detoxification process [64]. MDHA reductase (EC 1.6.5.4) reduces MDHA to ascorbate using NAD(P)H. The dehydroascorbate spontaneously produced by MDHA can be reduced to ascorbate by deshydroascorbate reductase (DHAR) using glutathione (GSH). During this process, glutathione is oxidized (GSSG). The cycle is completed when GR converts GSSG to GSH with the help of NAD(P)H as a reducing agent [65]. In citrus, the increase in specific SOD activity observed in triploids (3x) and Ellendale tangor genotypes is thought to be linked to low MDA levels and, consequently, to the limitation of the lipid peroxidation process [66]. In their study, Lourkisti et al. [66], reported that increased APX and CAT activities in most 3x citrus genotypes appeared to be sufficient to prevent oxidative damage by reducing H_2_O_2_ accumulation. In apple (Malus domestica) the gene MdTPR16 belongs to the TPR family, and promoter analysis shows that MdTPR16 contains multiple stress response elements, like the drought stress response elements [67]. MdTPR16 could trigger the up-regulates drought related genes in Arabidopsis and in apple. Under drought stress, overexpression of MdTPR16 was shown to reduces oxidative damage in cells by lowering electrolyte leakage, malondialdehyde (MDA) content, and the oxygen and hydrogen peroxide levels [67].

During drought stress, ABA also plays a crucial role in the reduction of oxidative stress induced by reactive oxygen species (ROS) by enhancing antioxidant defenses, modulating signaling pathways, and regulating stress-responsive gene expression. The overexpression of a previously mentioned gene, MdPYL9 was also found to reduce oxidative stress as transgenic apple trees presented a reduced amount of ROS along with a higher activity of antioxidant enzymes appearance of dwarfism, which is a water-stress tolerance strategy, and that overexpression of enzyme genes MdAAO3 and MdCYP707A and markers genes MdRD22 and MdRD29, correlate well with higher ABA levels in dwarfing plants underscore the complex and crucial role of ABA in enhancing drought tolerance, revealing various genetic and physiological adaptations that can be leveraged to improve crop resilience [68].

Secondary metabolites

Drought stress triggers secondary metabolite biosynthesis as an adaptive mechanism in plants, particularly in fruit crops. Secondary metabolites (SMs) such as phenolics, flavonoids, terpenoids, and alkaloids protect plants by (1) Scavenging reactive oxygen species (ROS), (2) Maintaining osmotic balance, (3) Strengthening plant cell walls and (4) Enhancing drought tolerance through hormonal and signaling pathways. Drought-induced secondary metabolite production is tightly controlled by hormones (ABA, JA, SA, and ET) signaling pathways. Drought stress activates specific transcription factors (TFs) that regulate SM biosynthetic genes. Extensive studies showed that drought stress triggers the biosynthesis of secondary metabolites (SMs). These small organic compounds were found to play a vital role in the drought resistance in fruit crops. For instance, melatonin (MT), acting as a powerful antioxidant, seems to play a role in preserving fruit crop from photosynthetic damages induced by drought. In apple [69] and kiwifruit (*Actinidia chinensis*) [70] melatonin (MT) improved chlorophyll content under water stress conditions. It is believed that MT increases the quantum yield of PSII in these fruit crop. In citrus, melatonin was shown to enhances the activity of SOD, CAT, and APX and the expression of genes encoding these antioxidant enzymes under drought [71]. It also increased the contents of phenolic and flavonoid compounds that have high antioxidant effect [72]. In red grapes, water deficit could result in wines with a higher concentration of volatiles coumponds [52]. Earlier studies showed increases in expression of the flavonol synthase (VviFLS) genes under water deficit [73] and flavonol concentration during berry ripening [74]. Terpenoids are present in essential oils [75]. These compounds play a protective role under drought conditions in woody plants [76]. In both white and red grapes grapevine, it has been shown that water deficit increased the concentration of terpene alcohols [73,77].

### 2.3. Stress Signals

#### 2.3.1. Phytohormones

Drought stress detection at the root level activates downstream signals that play a vital role in combating drought stress in fruit crops [78,79]. ABA biosynthesis and/or accumulation could limit the negative effect of drought stress on photosynthesis, assimilate translocation and hence growth [80,81]. In *Malus domestica* it has been shown that genes responsible for ABA synthesis (e.g., NCED3) where up-regulated under drought stress while genes responsible in ABA degradation (e.g., CYP707A1 and CYP707A2) were down-regulated [82]. ABA production could improve xylem water potential and plant water uptake [83]. It may also enable proline accumulation and soluble sugar content, crucial for adaptation in a saline environment [84]. Increasing ABA concentration in the root xylem enables the plant to close its stomata and thus limit water loss through the leaves [85,86]. Drought tolerance of *Malus domestica* apple trees was improved by the overexpression of the MhYTP1 related to ABA production, stomatal density and stomatal closure [87]. Moreover, ABA receptors, namely PYR/PYL MD06G1034000 and MD12G1178800 were found to be activated after PEG treatment (i.e., induced water-stress) during the development of adventitious apple roots protein phosphatases PP2Cs which in turn activate SnRK2 kinases. At the end of the chain following these activations, an efflux of anions and potassium anions causes a reduction in turgor pressure in the guard cells of the stomata, which leads to stomatal closure. The same study found a general upregulation of most ABA response factors (ABFs and SnRK2s) and higher ABA levels in adventitious roots during water-stress action of ABA production after the detection of a drought stress is stomatal closure to reduce transpiration and ensure a higher water-stress tolerance [88].

When soil osmotic pressure increases, the roots sense the soil water shortage and produce ABA. ABA is then transported through the xylem to the shoot, where, in response, guard cells increase their ABA content. This provokes stomatal closure and thereby reduces stomatal conductance (*g_s_*) and transpiration rate. Decreasing transpiration allows plant to better tolerate drought stress [28,72]. Calvez et al., 2022 [89] studied the response to water deficit in pot condition of Mexican Lime (ML) and Persian Lime (PL) grafted onto *Swingle citrumelo* 2x (Cit2x) and 4x (Cit4x) rootstocks. During water deficit, water consumption trees estimation showed that the PL/Cit2x combination lost water much faster than the others did (4th day), followed by ML/Cit2x (7th day). PL/Cit4x lost the same amount of water on 8th day and LM/Cit4x on 9th day. In stressed roots of ML/Cit2x and PL/Cit2x, NCED4 a well-known factor involved in ABA biosynthesis was over expressed, suggesting a possible ABA translocation to the scion part. Moreover, the expression of various genes related to ABA signaling such as “highly ABA-induced PP2C gene2” (Ciclev10023343m.g) showed significant upregulation in 4× *Swingle citrumelo*.

Water scarcity reduces the contents of cytokinins (CTK) and auxins (AUX/IAA), stimulating ethylene (ET) production. Salicylic acid (SA) production induce by drought stimulates the production of ROS to close stomata [78]. Under drought conditions, crosstalk between ABA and other hormones takes part in effective stress response. For instance, in apple under water stress, ABA and jasmonic acid (JA) have synergistic effects. Cytokinins (CKs) and auxin play a crucial role in plant drought adaptation by counteracting the effects of ABA. In model plants, Auxin and ABA could antagonistically regulate ascorbate (AsA) accumulation to scavenge ROS [90]. Cks role involves various physiological mechanisms, such as the specialization of vascular cells to improve root health and enhancing the communication network between different plant organs [91]. Salicylic acid (SA) is also known to stimulates the antioxidant Defense system and photosynthetic performance of *Aristotelia chilensis* plants [92].

Brassinosteroid hormone has been shown to antagonize the signal components of the ABA pathway to regulate the drought response [78]. Brassinosteroids (BR) act as a powerful growth inducer and aid the stress response [93,94]. They form a group of polyhydroxylated steroids that have been recognized as a class of phytohormones [45], which are essential for plant growth and development, as well as in abiotic stress responses [94]. They stimulate antioxidant activity [95], which allows for the mitigation of reactive oxygen species (ROS) under stress conditions. Application of BR in osmotic stress due to water deficit, has been shown to increase the content of osmolytes such as proline in peach [96] and apple [97].

#### 2.3.2. Transcription Factors

Drought stress signaling triggers downstream transcriptional regulatory responses. ABA-dependent and ABA-independent pathways for example activates various regulators of gene expression. Among them transcription factors (TFs) play a particularly important role in regulating the plant’s response to stress conditions. In apple MdWRKY115 TF binds to the MdRD22 promotor and increases drought tolerance [98]. ABSCISIC ACID-INSENSITIVE5 (MdABI5) is a TF in apple that activates the expression of the genes EARLY METHIONINE-LABELED6 (MdEM6) and RESPONSIVE TO DESICCATION 29A (MdRD29A) that respond to ABA and increase drought tolerance [99]. Drought stress in citrus trees activates the expression of CiBZR1, a member of the citrus BZR transcription factor family, that upregulates CiFRI (FRIGIDA) [100]. Overexpression of CiFRI in drought stressed transgenic lines, increases ROS scavenging [99,100]. The up regulated gene expression increases chlorophyll content level and improves apple growth in drought conditions. Although transcriptional regulation is commonly researched as a way for plants to tolerate abiotic stress, other molecular mechanisms can also be employed like post transcriptional regulations and epigenetic regulation on DNA level. In *Citrus sinensis*, water stress led to overexpression of CsMYB96 triggered by ABA signaling. CsMYB96 is a R2R3 MYB transcription factor. Once activated it binds to the promoters of different genes. CsPIPs (in citrus CsPIP1;1 and CsPIP2;4 code for aquaporins) and three wax-related genes (WRGs) such as ECERIFERUM1 (CsCER1) and beta-ketoacyl-CoA synthases (CsKCS4 and CsKCS12) where identified as CsMYB96 binding target. CsMYB96 can directly repress the expression of CsPIPs, preventing water loss. While it can activate the wax-related genes and promote wax biosynthesis, which also contribute to limit water loss [101]. In pear (*Pyrus betulaefolia*), PbERF3 transcription factor interaction with PbHsfC1a and activates the transcription of PbNCED4 and PbPIP1;4. These genes are involved in hydrogen peroxide transport and ABA biosynthesis [102]. Silencing PbERF3 resulted in reduced drought resistance. Thus, it is believed that this TF could play a crucial role in drought stress response. New research targets this regulatory module that plants use to combat drought stress. The aim of this ongoing work is to provide new insights into developing genetically modified crops with improved drought resistance. Beyond the physiological and biochemical adaptations, recent studies highlight the crucial role of transcriptional and epigenetic regulations in orchestrating fruit crops’ responses to water stress.

## 3. Epigenetic Regulations Involved in Water Stress Responses

Most of the physiological response described in this review is accompanied by a fine-tuned transcriptional response [97]. Epigenetics is the study of heritable modifications in gene expression not associated with changes in the gene sequence itself [98]. It involves structure and accessibility in transcription, without involving mutations or modifications in the sequence of DNA bases [99]. Cytosine methylation is the most studied epigenetic changes. It is generally associated with gene silencing and/or transcription, genome stability, genomic imprinting and transcription repression [100,101].

In annual plants, genome-wide studies showed changes in DNA methylation levels, including in the internal part of promoters of drought response genes [102,103,104,105,106]. These changes were specifically determined by the positive or negative regulation of the gene under drought stress. Outside fruit trees, water deficit also leads to changes in the methylation level of numerous genomic regions in poplar, some of which are associated with regions potentially directly related to plant adaptation [107]. Conversely, when the deposition of these modifications is affected, stress tolerance is reduced [108]. This is also the case in apple trees, where these changes affect genes that regulate phytohormones, but also genes directly involved in the deposition or deletion of cytosine methylation [109]. Global DNA methylation changes also accumulate in drought-stressed citrus trees [110,111], but their specific distribution need to be identified.

There is increasing evidence that plants exhibit dynamic methylation levels under drought stress. It is now known that ABA can act as a regulator of gene expression by inducing changes in methylation and histone acetylation, but also through the involvement of specific non-coding RNAs that allow the formation of a chromatin loop linking a specific locus to a distal enhancer [112,113]. ABA repressive activity could rely on DNA methylation of the promoter regions.

Chromatin remodeler can also play an important role in drought resistance. In apple, MdRAD5B is a chromatin remodeler induced by drought-stress. It modulates the polycomb-mediated trimethylation at position K27 of the histone 3 (H3K27me3), through the degradation of mdLHP1 that protects H3K27me3 on silent genes [114]. In consequence, drought-induced activation of MdRAD5B leads to the activation of up to 600 genes during drought stress, participating in drought resilience of the tree.

Plant performance to resist to an abiotic stress is increased after a first exposure to a transient, non-lethal stress. This phenomenon is known as priming. Epigenetic modifications in DNA and associated histone proteins have been shown to encode short-term and long-term memory within the same plant or facilitate transgenerational effects [115]. Further studies, specifically dedicated to the study of these changes in fruit trees, will be needed to confirm these initial observations. But this line of research is very important for determining whether the memory of stress can be passed from one generation to another by these epigenetic modifications. Although certain modifications associated with histones also appear to play a role in stress memory [116], cytosine methylation remains the modification most stably perpetuated to subsequent generations in plants, coupled to replication or via small RNAs [117,118]. Little work has been done on the perpetuation of changes over several generations, and the initial responses obtained in *A. thaliana* need to be confirmed [119]. Potentially, combining site-directed epigenetic manipulation with specific promoters opens up new opportunities to test and engineer spatiotemporal patterns of priming [120].

### 3.1. A Potential Role of Transposable Elements in Plant Adaptation

Cytosine methylation is also largely involved in the transcriptional repression of transposable elements (TEs), which make up a large proportion of many plant genomes and play a role in their evolution [103]. TEs are mobile genetic parasites that can move and replicate within a genome, affecting genome structure and function (Figure 2A) [104].

Upon specific developmental or environmental cues, discrete families of TEs are activated and transposed to new genomic loci, leading to changes in gene activity, including gene knockout, truncation and altered expression. Therefore, TE neo-insertion often results in deleterious defects for the host organism, and host plants have evolved sophisticated regulatory pathways to avoid TE activation through multiple layers of epigenetic regulation, notably involving DNA methylation, histone modifications and post-transcriptional gene silencing [105]. However, TE-induced phenotypic variation has recently been essential for the domestication and improvement of several major crops [106,107], highlighting that TEs induce desirable phenotypic change in agricultural environments.

Crops with robust phenotypes are required in agriculture, but the presence of a TE in the promoter region of a gene sometimes lead to different transcriptional output. However, some TEs are induced by environmental cues such as drought, heat stress or cold [108,109,110,111,112], facilitating, the accumulation of genomic variation and potentially phenotypic variability. If such phenotypic variability is important to breeders, such variability is usually feared by the farmers. A striking example is the blood orange phenotype obtained after the neo-insertion of a Copia-like retrotransposon adjacent to the Ruby encoding gene, a positive regulator of anthocyanin production (Figure 2F) [109]. Because the LTR transcriptional activation is dependent of cold, the accumulation of anthocyanin is also cold-dependent. Phenotypic variability can also be due to differential epigenetic status, in oil palm trees, where two KARMA epialleles can be found, methylated or unmethylated, and lead to a normal or mantled fruit phenotypes respectively (Figure 2E) [113].

More specifically, drought-stress can provoke an increase TE activation, as shown in Coffea where TE transcripts accumulate differentially depending on the cultivars, with a stronger accumulation detected in drought tolerant *C. arabica* and *C. canephora* cultivars compare to drought sensitive *C. arabica* cultivars [115], although we don’t know whether this increase is linked to the tolerance status in that case.

The potential impact of MITE activity in fruit tree drought adaptation was recently demonstrated in apple trees where a newly inserted MITE modified the transcriptional activity of MdRFNR1 (Figure 2D) [116]. MdRFNR1 is a NADP^+^ oxidoreductase implicated in the modulation of the redox system and one allelic version contains a PIF/Harbinger MITE in his promoter, which becomes methylated. Interestingly, the consequence of the transcriptional activity of this MdRFNR1 is actually increased due to the recruitment of anti-silencing factors including a co-chaperon DNAJ protein and in particular SUVH1 and 3-like histone methyltransferase, whose activity enhance transcription [117]. This example also highlights how mobile-DNA induced genetic changes affect the epigenetic status of genes.

Although only a handful of examples were described so far, more systematic analysis should reveal their impact, especially since TE are the source of the majority the extensive genomic variability that exist among the different fruit tree varieties as shown in apple [118].

### 3.2. mi-RNA

Amongst mobile signals, small RNAs played a significant role in rootstock’s impact on scion tolerance through epigenetic mechanisms. miRNAs are generated by DICER-LIKE1 (DCL1) enzymes from stem-loop precursors encoded by endogenous *MIR* genes [119,120]. Their regulation activity involves RNA interference (RNAi) machinery by binding to the untranslated regions (UTRs) of mRNA to suppress protein translation or decay mRNA [121,122,123]. Upon exposure to abiotic stresses, particularly drought and salinity, plant miRNAs can act as mediators in the regulation of molecular signaling cascades [122]. Stress-responsive regulatory sequences can be located within the *MIR* gene promoters [121]. In Vitis, a study investigated drought stress effects on microRNA (miRNA) abundance in a drought-resistant grapevine rootstock, M4 (*Vitis vinifera × Vitis berlandieri*), compared with a commercial cultivar, Cabernet Sauvignon [120]. miRNA abundance differed depending on the genotype under water deficit conditions. Drought-responsive miRNAs concentration was affected by reciprocal grafting. This result suggests that either the movement of signals inducing miRNA expression in the graft partner or, possibly, miRNA transport between scion and rootstock.

## 4. Trends in Genetic Research and Breeding Programs to Cope with Drought Stress in Crops

### 4.1. Genome-Wide Association Studies (GWAS) and Epigenome-Wide Association Studies (EWAS)

To date in plants, only a few studies investigate alleles whose effects are modulated by environmental conditions, such as abiotic stress. The existing study focus on model plant *Arabidopsis thaliana* [124,125] or cereals [126,127]. Up to now, the identification of the genetic determinants of drought response originating from a large diversity of fruit crops species remains limited [128]. QTL classification according to the prevalence of their effect under the different conditions (QTL by environment interactions) is commonly performed using two complementary approaches. In multivariate QTL mapping models, the effects of a given QTL are assessed across the environmental conditions. The other approach aims to build composite variables by measuring phenotypic plasticity and univariate mapping models. In crop plants, the availability of a high-throughput genotyping assay, makes it possible to assess this classification via conventional linkage mapping [129,130,131]. Genome-wide association studies (GWASs) are also increasingly used. Their advantage over linkage mapping is the possibility to explore the genetic diversity and the numerous recombination events present in germplasm collections. GWAS allows higher resolution mapping if the LD (linkage disequilibrium) is low enough in the population [132].

Considering the number of molecular actors activated after the recognition of a drought stress signal, mass analysis techniques seem to be a great tool to deepen the research on this subject. Genome-wide association studies (GWAS) aim to identify associations of genotypes with phenotypes by testing for differences in the allele frequency of genetic variants between individuals who are ancestrally similar but differ phenotypically [132]. This method simultaneously tests thousands of genetic variants across genomes to identify those linked to the studied trait. GWAS is a powerful tool for investigating complex traits related to environmental stress. Studies in plants have revealed genes and loci linked to abiotic stresses, including water stress, offering valuable insights for breeding and developing future crops [133].

To further accelerate molecular breeding programs for crop improvement it appears important to consider that specific variation in DNA sequence is not the only determinant of plant stress adaptation [134]. The different epigenetic signatures can be used as a form of variation that impact the chromatin structure and affect transcription factors access to modulate gene expression [135]. Study of methyl quantitative trait loci (QTL) or EGWAS, is possible by using high-throughput sequencing methods. This method is considered as a powerful research tool for displaying the systematic associations of genetic and epigenetic variations, especially in terms of cytosine methylation onto a given genomic region [136]. This technology could be a potential research tool for deciphering correlations between genotype to epigenotype and to phenotype in crop plant populations [134].

In addition, structural variation (SV)mediated by TE neo-insertion can be rapidly accumulate in plant affected in epigenetic regulators or in genome stability [137,138]. Widely used in crops today, screening for changes in natural variation in fruit trees will undoubtedly reveal specific variation associated with better drought tolerance, possibly using strategies using GWAS that focus on epigenetic variation (Epi-GWAS) [139] or TE mobility via the identification of TE Insertion Site (TIS or TE -GWAS) [140]. Indeed, among more than 142 thousands SV found from 19 assembled poplar genomes, an important part was generated by TE. As an example, a MITE insertion in the CUC2 gene was shown to be involved in leaf serration [141]. Moreover, to assess globally the impact of SV, generated or unliked by TE, on a given phenotype, SV-GWAS is a very powerful approach [142]. Finally, with the advent of GWAS methodologies capable of capturing heritable epistatic interactions—referred to as Next-Generation GWAS [143] the analysis of current and upcoming datasets opens exciting new pathways for genetic research.

### 4.2. Polyploidy

Increasing evidence in the growing pool of sequenced genome tend to demonstrate the common occurrence of polyploidy in crops. This includes for example, fruit crops like citrus [144], oilseed rape or canola (*Brassica napus L.*) [145] and banana (*Musa acuminata L.*) [146]. While the production of triploid plants (3n) enables aspermic varieties to be obtained, and therefore seedless fruit, tetraploid plants (4n) are recognized for their tolerance to biotic and abiotic constraints, qualities sought after for the selection of new rootstocks [147,148]. Polyploid plants are known to have large cell size and more massive organs (such as roots, leaves, flowers, and seeds) than their diploid counterparts [144]. Tetraploids citrus for instance are characterized by stunted growth compared to their diploid counterparts. But despite this growth delay, their cells and organs are larger and more massive. At leaf level, tetraploids have larger stomata but at lower densities than diploids. Tetraploid citrus roots are shorter, thicker, less branched and probably have lower hydraulic conductance than their diploid counterparts. These histological and morphological traits could lead to better adaptation to water deficit and salt stress. It has been demonstrated that tetraploid lemon (*C. limonia Osb.*) [149], as tetraploid Rangpur lime citrus [19] could have better drought-tolerance ability than their diploids counterparts. Drought tolerance in diploid (2x) and autotetraploid (4x) clones of Rangpur lime (*Citrus limonia*) rootstocks grafted with 2x Valencia Delta sweet orange (*Citrus sinensis*) scions, named V/2xRL and V/4xRL was investigated by Allario et al. (2013) [19]. In roots, ABA content was higher in V/4xRL and was associated to a greater expression of drought responsive genes, including CsNCED1, a pivotal regulatory gene of ABA biosynthesis. This study suggested that tetraploidy modifies the expression of genes in Rangpur lime citrus roots to regulate long-distance ABA signaling and adaptation to stress.

Multiple studies also highlight the genomic consequences and underlying mechanisms for generating evolutionary novelty and morphological diversity [150,151,152,153,154,155,156]. In addition to the genetic changes, polyploidy has been described has the driving force behind a multitude of epigenetic modifications [157]. This includes DNA methylation, histone modifications, and chromatin remodeling [158,159,160]. When polyploidization results from the fusion of at least one unreduced (2n) male or female gamete following various meiosis abnormalities, it is referred to as sexual polyploidization. Allotetraploid genotypes result from the hybridization of two diploid gametes that have not undergone meiotic reduction. Gene expression changes in an allotetraploid individual can be additive or nonadditive (deviated from mid-parent value, MPV), suggesting an epigenetic cause [161].

Polyploidization, often linked to suppressed silencing mechanisms, may also promote TE activation [162]. Under drought stress, a synergistic effect could arise, further amplifying TE activation. In autopolyploids, genome duplication can alter dosage regulation on biological networks. In allopolyploids, interspecific hybridization could induce genetic and epigenetic changes. Ploidy level influence could amplify theses effects [163]. Some of this important genetic and epigenetic changes can be stabilized and transmitted as epialleles into the progeny. Thus, they are subject to natural selection and can contribute to plant evolution and crop domestication. Understanding, DNA methylation and chromatin modifications influence on gene expression and phenotypic variation in plant polyploids is a promising open field of research. This work could have concrete implications for growers.

### 4.3. Grafting

To cope with biotic and abiotic constraints, some fruits crop (grapevine or citrus for example) are grafted onto rootstocks selected, among other things, for their adaptive properties [19,53]. A rootstock is a woody or herbaceous plant stripped of its aerial part and provided with a root system on which a graft is implanted. Rootstock and scion genotypes can confer to the plant traits of drought adaptability influencing respectively the capacity of water extraction from the soil and the sensitivity of the stomatal control [17,52,53]. Rootstocks play then a major role to water stress tolerance by controlling and adjusting the water supply to shoot transpiration demand. Moreover, root signals move though xylem to shoots and affected the shoot physiological metabolism [15]. Phytohormones such as ABA, JA, SA, and ET are central player in the root to shoot communication signaling in environmental stresses response [53,164]. Long-distance signaling in root-to-shoot communication can involve chemical signals and water status which was also called as hydraulic signal [82,165]. Among chemical signals, abscisic acid (ABA), pH, cytokinins, ethylene precursors, malate can be implicated in root to shoot communication under drought [166]. Rootstocks are selected above all for their resistance to disease, pests, climate and soil type. For example, in grapevine M4 [(*Vitis vinifera × V. berlandieri) × V. berlandieri × cv Resseguier n. 1*], can be selected for its high tolerance to water deficit and salt exposure [51]. In citrus, *Cleopatra mandarin* rootstock also shows good tolerance to cold, salt stress, drought and limestone. However, it is susceptible to tristeza, exocortis and phytophtora. *Poncirus trifoliata* is characterized by a superficial root system, making it sensitive to drought, acidity and alkalinity. *Poncirus trifoliata* are characterized by a superficial root system, making them sensitive to drought, acidity and alkalinity. Poncirus hybrids such as citrumelo swingle 4475 [167] (*Poncirus trifoliata × C. paradisi*) and citrange Carizzo (*Poncirus trifoliata × C. sinensis*) are also commonly used as rootstocks and inherited from their Poncirus parent a good tolerance to tristeza but a sensitivity to chlorides and drought. Other rootstock such as Rangpur lime (*C. limonia*) and sour orange (*C. aurantium*) are known to be more tolerant of salt stress and drought. On the other hand, lime is susceptible to phytophtora and exocortis, unlike sour orange.

Grafting can induce significant epigenetic changes, such as DNA methylation and histone modifications, affecting the expression of genes linked to drought tolerance. For example, studies on self-grafted tomato plants have revealed epigenetic changes that improve resilience to water stress [168]. Research has shown that small RNAs, such as siRNAs, can be transferred between rootstock and scion, leading to epigenetic modifications in recipient tissues [169]. This mechanism may influence tolerance to water stress by regulating the expression of specific genes. Studies on citrus grafting combinations have shown that repeated exposure to water deficit can result in stable epigenetic modifications, such as changes in DNA methylation, which confer stress memory and improve drought tolerance [170].

### 4.4. Genetic Editing Techniques

World’s population is expected to reach 10 billion by 2050 [171], meanwhile, available farmland and water are decreasing annually. Transgenic technologies are thought to be fastest way to improve plant drought tolerance [172]. Genome editing techniques are believed to be the most recent approach to genetic engineering. These techniques include for instance zinc-finger nucleases, transcription activator-like effector nucleases, and the Clustered Regularly Interspaced Short Palindromic Repeats (CRISPR)/CRISPR-associated protein 9 (Cas9) system. They provide precise genetic modifications are performed at specific sites [173].

Studies have shown how the CRISPR/Cas9 system can improve drought tolerance in plants by targeting specific genes [174]. Most research has been carried out on model plants such as Arabidopsis or rice. For example, modification of the OST2 gene in Arabidopsis improved stomatal closure, thereby reducing water loss [175]. In addition, activation of the AREB1 gene, an abscisic acid (ABA)-responsive transcription factor, enhanced plant response to water stress by increasing the expression of genes such as RD29A [176]. Similar approaches could be applied in cultivated plants. For example, editing of the NPR1 gene in tomato showed a reduction in drought tolerance, indicating that this gene plays a crucial role in the response to water stress. Transgenic plum lines were developed by overexpressing antioxidant enzymes such as superoxide dismutase (SOD) and ascorbate peroxidase (APX), derived from spinach and pea respectively. These modifications improved tolerance to salt and water stress, by modulating enzymatic and non-enzymatic antioxidant systems such as glutathione and ascorbate. A transgenic line with high APX activity showed tolerance to severe water stress [177].

Several studies are also underway to apply epigenomic editing techniques, such as CRISPR/dCas9, to modulate the expression of genes linked to drought tolerance with-out altering the DNA sequence. This approach offers the potential to develop fruit trees that are more resilient to water stress.

## 5. Conclusions

Drought stress affects numerous aspects of fruit crops biology, and the plants have evolved to a diverse array of adaptive defense mechanisms, including physiological adjustments, hormonal changes, genetic and epigenetic alterations, to enhance their tolerance under water scarcity. Key physiological responses include reduced leaf water potential, lower turgor pressure, stomatal closure, altering root xylem formation and employing antioxidant defenses. Development of drought-resistant crop varieties by combining sequencing approaches and genetic engineering with epigenetic studies (i.e., turn the loci into a sustained methylated or demethylated state) could help building new germplasm library with enhanced drought resistance. As shown in A. thaliana, epigenetic editing that influences gene expression without altering the genetic code can be useful to activate or repress drought-responsive genes, providing another layer of regulation for drought tolerance [178].

As studies have demonstrated the transmission of stress memory across generations, further research into transgenerational epigenetic inheritance could open new avenues for breeding programs. High-throughput phenotyping combined with systematic environmental characterization can also be used. This strategy is intended to provide direct insights into dynamic traits that affect physiological variables and fruit quality on an orchard scale. By merging genomic data with machine learning techniques, the goal is to predict and determine optimal plant types for different conditions, including drought stress tolerance. This can lead to a better understanding of improved breeding and domestication for more climate tolerant cultivars of fruit trees crops. Exploring the interactions between scion/rootstock combination and epigenetic marks can lead to improved crop resilience and food security.

## Figures and Tables

**Figure 2 plants-14-01769-f002:**
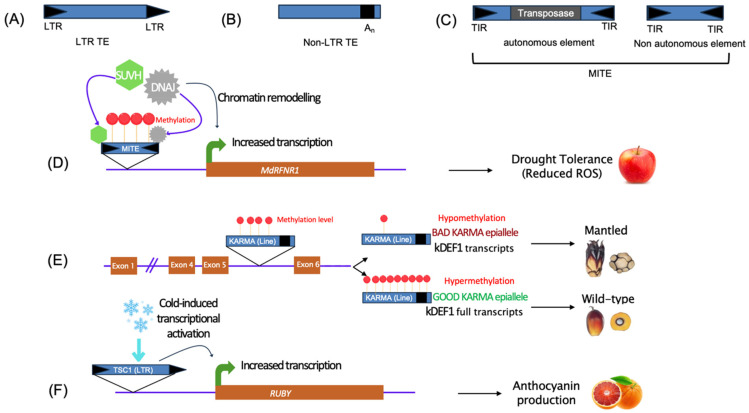
Role of transposable elements (TEs) in phenotypic changes and stress responses in fruit trees Structure of TE with LTR (**A**), non-LTR (**B**) and of the autonomous and non-autonomous MITE (**C**). Insertion of a MITE element in the promoter of the MdRNFR1 gene affect its epigenetic and transcriptional status in Apple tree, inducing a reduced ROS accumulation during drought stress [114] (**D**). Insertion of non-LTR in the intron of the DEF1 gene provoke differential methylation allelic version that impact its transcriptional status and lead to the Mantled fruit phenotype in palm oil [113] (**E**). Insertion of the TSC1 LTR TE in the promoter of the RUBY gene affect provoke its transcriptional activation under cold stress, provoking the accumulation of anthocyanin in the orange fruit [109] (**F**).

## Data Availability

No new data were created or analyzed in this study. Data sharing is not applicable to this article.

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
