# Peer review of "Physiologic, Genetic and Epigenetic Determinants of Water Deficit Tolerance in Fruit Trees"

_plants, 2025, doi:10.3390/plants14121769_

Round 1
Reviewer 1 Report
Comments and Suggestions for Authors
This review summarizes the various mechanisms in response to water deficits in fruit crops, encompassing physiological, genetic, and epigenetic stress responses. The structure of the manuscript is well-organized and provides a comprehensive understanding of water stress in fruit crops. My recommendation is acceptance with minor revisions.
Figure 1 is too small and indistinct. I suggest enlarging and presenting the figure separately.
At line 396, why is the section “3.2 mi-RNA” placed after section 2.3.2? It should be included in the section “3. Epigenetic Regulations Involved in Water Stress Responses.”
In section 4.3, add some examples of the applications of genetic editing techniques in fruit trees.
At line 626, there is an additional dot at the beginning of the paragraph. Is a sentence missing here?
Author Response
This review summarizes the various mechanisms in response to water deficits in fruit crops, encompassing physiological, genetic, and epigenetic stress responses. The structure of the manuscript is well-organized and provides a comprehensive understanding of water stress in fruit crops. My recommendation is acceptance with minor revisions.
Comment 1: Figure 1 is too small and indistinct. I suggest enlarging and presenting the figure separately.
Response 1: As suggested, we enlarged figure 1.
Comment 2: At line 396, why is the section “3.2 mi-RNA” placed after section 2.3.2? It should be included in the section “3. Epigenetic Regulations Involved in Water Stress Responses.”
Response 2: As suggested, the section 3.2 on “mi-RNA” has been moved to section 3 line 476.
Comment 3: In section 4.3, add some examples of the applications of genetic editing techniques in fruit trees.
Response 3: As suggested, we added examples of editing techniques in fruit such as grafting (section 4.3 lines 637 to 648) and genetic editing via the CRISPR Cas9 system (section 4.4 lines 660 to 682).
Comment 4: At line 626, there is an additional dot at the beginning of the paragraph. Is a sentence missing here?
Response 4: Additional dot has been removed.
Reviewer 2 Report
Comments and Suggestions for Authors
The great value of this manuscript is that there are very few works about genetics and EPIGENETICS in fruit tree since it is very difficult to study the argument. This review manuscript provides a comprehensive overview of the complex mechanisms underlying water deficit tolerance in fruit trees. The authors cover a wide array of topics, from physiological and biochemical responses to genetic and epigenetic determinants, and also touch upon modern breeding strategies. The paper is well-organized, references are sufficient, and a good accent has been posed on climate change and fruit crop production. The central question can be identified in, "how can we move towards a more integrative definition (physiological, biochemical, genomical and epigenomical) of fruits crops drought tolerance?", is well addressed and guides the review effectively. The manuscript is a good contribution to this research field.
Here some suggestions:
Minor Points for Revision:
- Nomenclature/Typographical Errors:
- In the introduction, when defining "major" fruit crops, "sherry trees" is mentioned. This is likely a typographical error and should probably be "cherry trees." Please verify and correct.
- The section on "Genetic editing techniques" is numbered "4.3" but should follow section 4.3 (Grafting) and thus be numbered "4.4".
- Structural Cohesion of Section 3 (Epigenetics and miRNA):
- Section 3 is titled "Epigenetic regulations involved in water stress responses". Section 3.1 discusses transposable elements, which is fitting.
- However, Section 3.2 is "mi-RNA". While miRNAs can be involved in epigenetic pathways (e.g., RNA-directed DNA methylation), their primary role is post-transcriptional gene regulation. The current text describes their function as mobile signals and regulators of mRNA but doesn't strongly link them to heritable epigenetic modifications in the context presented.
Suggestion: Either explicitly elaborate on the role of these miRNAs in inducing heritable epigenetic changes relevant to drought in fruit trees (if strong evidence exists) or consider repositioning/rephrasing this subsection. For instance, it could be part of a broader section on "Regulatory RNAs in stress response" or the authors could clarify that "epigenetic" is used in a broader sense here to include such regulatory networks.
- Clarity of Section Title 2.2.1:
- The title "Genes related to primary and secondary metabolism" for section 2.2.1 is very broad. While the content (Photosynthesis, Cellular water regulation, Osmoprotectants, Antioxidants, Secondary metabolites) does fall under this umbrella, the subsections are more process/compound-oriented. The current title is not incorrect, but a more direct title reflecting the content structure might improve clarity for the reader, e.g., "Molecular and Metabolic Adjustments to Water Scarcity." This is a minor suggestion for the authors' consideration.
- Minor Textual Refinements:
- Page 1, Abstract: "Water availability is now recognized as one of the most important environmental factors limiting the growth and productivity of fruit crops." Consider rephrasing slightly for conciseness, e.g., "Water availability is a primary environmental factor limiting fruit crop growth and productivity."
- Page 1, Introduction: "The growth stages of fruit trees are very sensitive to temperature and irrigation." While true, the focus of the review is drought. Perhaps frame it more directly towards water availability, e.g., "Fruit tree growth stages are highly sensitive to water availability and temperature."
Author Response
Here some suggestions:
Minor Points for Revision:
Comment 1 : Nomenclature/Typographical Errors:
- In the introduction, when defining "major" fruit crops, "sherry trees" is mentioned. This is likely a typographical error and should probably be "cherry trees." Please verify and correct.  
Response 1 : Sorry for this mistake. The typographical error has been corrected.
- The section on "Genetic editing techniques" is numbered "4.3" but should follow section 4.3 (Grafting) and thus be numbered "4.4".       
Sorry for this mistake. Thenumbering has been corrected.
                                          
Comment 2: Structural Cohesion of Section 3 (Epigenetics and miRNA):
- Section 3 is titled "Epigenetic regulations involved in water stress responses". Section 3.1 discusses transposable elements, which is fitting.  
- However, Section 3.2 is "mi-RNA". While miRNAs can be involved in epigenetic pathways (e.g., RNA-directed DNA methylation), their primary role is post-transcriptional gene regulation. The current text describes their function as mobile signals and regulators of mRNA but doesn't strongly link them to heritable epigenetic modifications in the context presented.    
Suggestion: Either explicitly elaborate on the role of these miRNAs in inducing heritable epigenetic changes relevant to drought in fruit trees (if strong evidence exists) or consider repositioning/rephrasing this subsection. For instance, it could be part of a broader section on "Regulatory RNAs in stress response" or the authors could clarify that "epigenetic" is used in a broader sense here to include such regulatory networks.
Response 2: The section 3.2 on “mi-RNA” has been moved to section 3 line 476.
Comment 3 : Clarity of Section Title 2.2.1:
- The title "Genes related to primary and secondary metabolism" for section 2.2.1 is very broad. While the content (Photosynthesis, Cellular water regulation, Osmoprotectants, Antioxidants, Secondary metabolites) does fall under this umbrella, the subsections are more process/compound-oriented. The current title is not incorrect, but a more direct title reflecting the content structure might improve clarity for the reader, e.g., "Molecular and Metabolic Adjustments to Water Scarcity." This is a minor suggestion for the authors' consideration.  
Response 3: Thank you for this proposition. We have renamed the title 2.2.1 to “Molecular and Metabolic Adjustments to Water Scarcity”. line 86.
Comment 4 : Minor Textual Refinements:
Page 1, Abstract: "Water availability is now recognized as one of the most important environmental factors limiting the growth and productivity of fruit crops." Consider rephrasing slightly for conciseness, e.g., "Water availability is a primary environmental factor limiting fruit crop growth and productivity."  
Response 4: Thank you for this proposition. We rephrased this sentence line 23-24.
Comment 5: Page 1, Introduction: "The growth stages of fruit trees are very sensitive to temperature and irrigation." While true, the focus of the review is drought. Perhaps frame it more directly towards water availability, e.g., "Fruit tree growth stages are highly sensitive to water availability and temperature."
Response 5: Thank you for this proposition. We performed the correction line 39-40.
Reviewer 3 Report
Comments and Suggestions for Authors
Author Response
Comment 1: In Figure1, the illustration is not enough, should contain morphological, physiological and transcriptional response in order.
Response 1: Thank you for this suggestion. We now added some aspect of the physiological ant transcriptional response to drought in Figure 1.
Comment 2: Only 2.2.1 in 2.2., the structure should be adjusted.
Response 2: We adjusted the structure (Line 86).
Comment 3: The transition between physiological responses (Section 2) and genetic/epigenetic mechanisms (Section 3) is abrupt
Response 3: Thank you for this comment. Indeed, a transition was lacking. We now addes the following sentence : “Beyond the physiological and biochemical adaptations, recent studies highlight the crucial role of transcriptional and epigenetic regulations in orchestrating fruit crops' responses to water stress” (lines 396-398).
Comment 4: Ensure consistent formatting (e.g., some in-text citations lack year; reference list includes placeholder citations like “Akakpo et al., 2020”).
Response 4: We performed the suggested corrections (line 554, 473 and in the legend of Figure 2).
Comment 5: The type writing needs carefully checked.
Response 5: Typographic errors have been corrected like “cherry trees” line 52.